# Anticancer Efficacy of Nonthermal Plasma Therapy Combined with PD-L1 Antibody Conjugated Gold Nanoparticles on Oral Squamous Cell Carcinoma

Jinyoung Park [1], Yoon-Seo Jang [1,2], Jeong-Hae Choi [2,3], Miheon Ryu [4], Gyoo-Cheon Kim [3], June-Ho Byun [5], Dae-Seok Hwang [1,*,†] and Uk-Kyu Kim [1,*,†]

[1] Department of Oral and Maxillofacial Surgery, School of Dentistry, Pusan National University, Yangsan 50612, Korea; forfind@pusan.ac.kr (J.P.); saysay282@nate.com (Y.-S.J.)

[2] Feagle Co., Ltd., Yangsan 50614, Korea; monday27@pusan.ac.kr

[3] Department of Oral Anatomy, School of Dentistry, Pusan National University, Yangsan 50612, Korea; ki91000m@pusan.ac.kr

[4] Department of Oral Pathology, School of Dentistry, Pusan National University, Yangsan 50612, Korea; apollon@pusan.ac.kr

[5] Department of Oral and Maxillofacial Surgery, School of Medicine and Hospital, Gyeongsang National University, Jinju-si 52727, Korea; surbyun@gnu.ac.kr

[*] Correspondence: dshwang@pusan.ac.kr (D.-S.H.); kuksjs@pusan.ac.kr (U.-K.K.)

[†] Uk-Kyu Kim and Dae-Seok Hwang have equally contributed to this work and should be considered co-corresponding authors.

**Abstract:** Combination therapies for the treatment of oral squamous cell carcinoma have been studied extensively and represent a synergistic approach with better outcomes than monotherapy. In this study, a novel combination therapy was investigated using gold nanoparticles (GNP) conjugated to programmed cell death protein ligand 1 (PD-L1) antibodies and nonthermal plasma (NTP). The present study describes the effectiveness of NTP using PD-L1 antibody conjugated to GNP in PD-L1 expressing SCC-25 cells, an oral squamous cell carcinoma line. Immunocytochemistry revealed higher levels of PD-L1 expression and an increase in the selective uptake of PD-L1 antibody + GNP on SCC-25 cells compared to HaCaT cells. In addition, cell viability analyses confirmed higher levels of cell death of SCC-25 cells after treatment with PD-L1 antibody, GNP, and NTP compared to HaCaT cells. Among the experimental groups, the highest cell death was observed upon treatment with PD-L1 antibody + GNP + NTP. Following the Western blot analysis and immunofluorescence staining, the expression of apoptosis-related proteins was found to increase after treatment with PD-L1 antibody + GNP + NTP among the other experimental groups. In conclusion, the treatment of SCC-25 cells with PD-L1 antibody + GNP + NTP significantly increased the number of dead cells compared to other experimental groups. The results of this in vitro study confirmed the therapeutic effects of PD-L1 antibody + GNP + NTP treatment on oral squamous cell carcinoma.

**Keywords:** gold nanoparticle; nonthermal plasma; oral cancer; PD-L1; combination therapy

## 1. Introduction

Despite the significant advances in cancer prevention, diagnosis, and treatment over the past 50 years, the prognosis of oral cancer has not improved. In South Korea, approximately 3600 people are diagnosed with oral cancer, resulting in approximately 1200 deaths, annually [1,2]. In addition, the surgical treatment of oral cancer often results in defects in major structures of the head and neck, causing both aesthetic and functional damage that can result in psychological disorders and reduced quality of life. Furthermore, cytotoxic chemotherapy and radiation therapy inevitably sacrifice normal cell and may cause side effects such as vomiting, nausea, diarrhea, hair loss, trismus, and inflammation [3]. To reduce these side effects, studies are actively being conducted with the aim of developing treatments that act solely on cancer cells.

Among these, a recent study investigated the use of monoclonal antibodies (mAb) to selectively attack cancer cells. In fact, mAb against cancer cell-specific antigens have been proposed as alternatives to cancer-selective treatment. However, most mAb acting alone do not show a sufficiently strong therapeutic effect. Nevertheless, cancer cell-specific antibodies continue to show beneficial effects when used in cancer treatments. Since these antibodies specifically bind to the surfaces of cancer cells, they show clinical potentials of cancer cell selectivity that are differentiated from conventional chemotherapy.

To find antigens that selectively target cancer cells, these antigens must be specifically expressed on cancer cells. In a previous study, PD-L1 was found to be highly expressed in patients with oral cancer and was associated with a poor prognosis [4]. PD-L1 was recently proposed as an immune checkpoint because of its association with the immune evasion of cancer. Moreover, the inhibition of PD-L1 has been shown to promote T cells with anticancer activity [5]. In fact, the achievement of the immune checkpoint block treatment in patients with various malignant tumors is changing the paradigm of the treatment [3,6]. Advances in immunotherapy improved the survival rates of cancer patients. In many clinical trials, monoclonal antibodies that target immune checkpoint proteins have demonstrated significant responses in the treatment of lung cancer and terminal stage melanoma patients [7]. The United States Food and Drug Administration (FDA) has accepted the use of three PD-L1 inhibitors and two PD-1 inhibitors for the cure of melanoma, lymphoma, and lung cancers [7,8].

However, a large number of patients do not react to immune checkpoint inhibitors or develop a resistance to treatment [9]. Therefore, the complete elimination of cancer and the reduction of its recurrence rate are difficult to achieve using only one method, such as immunotherapy [10]. Thus, many clinical trials involving immune checkpoint inhibitors are currently being conducted in combination with other treatment methods, including radiation therapy or chemotherapy, to improve the cancer survival rate and the synergistic effects on the suppression of the growth of cancer cells [8,10–12]. However, this approach has limitations in clinical practice due to its potential toxicity, resulting in dose limitations and low responses in patients. In this regard, if immunotherapy is combined with anticancer nanomedicine, it may be able to safely and effectively enhance the anticancer immune response, as well as increase the sensitivity of patients' responses to immunotherapy [13,14]. There are two important goals of developing cancer treatments: improved selectivity and enhanced cancer cell destruction. With the recent development of nanotechnology, gold nanoparticles (GNP) are being used for cancer treatment, as this material shows potential for achieving these two goals [13,15].

The combination of antibodies and nanomaterials has been found to have potential for selectively targeting cancer cells. In addition, the treatments using GNP have shown promising outcomes in the early clinical trials. GNP can also be used for photothermal therapy due to their high atomic number [16]. In a previous study, gold photothermal therapy was found to increase both the B- and T-cell populations while decreasing the myeloid-derived suppressor cells of immunosuppression, indicating that synergistic control is possible not only in the primary site but, also, in untreated distal tumors [17–19].

Within this context, we investigated whether nonthermal plasma (NTP) can amplify the anticancer effects of GNP conjugated to the antibody. NTP itself works effectively on head and neck squamous cell carcinoma cells [20], and recently, it has been reported that it has a remission effect when applied clinically [21]. Additionally, previous studies have attempted to create a cooperative anticancer effect on cancer cells by combining GNP with NTP [22–24]. As a result, it was confirmed that the cell cycle of cancer cells was stopped, and cell death was induced. These results indicate that NTP is an ideal auxiliary anticancer treatment method, whose combination with other treatments (chemotherapy and nanoparticle therapy) may regulate the microenvironment of cancer cells and inhibit cancer cell growth via cooperative action [25,26]. Here, NTP was used to amplify the effect of the GNP conjugated to the PD-L1 antibody to induce selective action. The results

presented in this study suggest that the use of combination therapy is an effective treatment for the treatment of advanced oral cancer.

## 2. Materials and Methods

### 2.1. Reagents

Gold nanoparticles (30-nm gold colloid) were purchased from BBI solutions (Cardiff, UK). Monoclonal mouse antihuman GAPDH, PARP, cleaved caspase-3, apoptosis-inducing factor (AIF), cytochrome C (cyt C), and β-actin antibodies were purchased from Santa Cruz Biotechnology (Dallas, TX, USA). The polyclonal rabbit antihuman PD-L1 antibody, 5% bovine serum albumin (BSA), and Nunc™ Glass Bottom Dish (diameter 40 mm) were purchased from Thermo Fisher Scientific (Waltham, MA, USA). Dulbecco's modified Eagle's medium and F-12 (DMEM/F-12), penicillin/streptomycin (100 µg/mL), and fetal bovine serum (FBS) were purchased from Gibco (Brooklyn, NY, USA). 11-mercaptoundecanoic acid (11-MUA), ethyl-dimethylaminopropyl carbodiimide (EDC), N-hydroxy succinimide (NHS), sulforhodamine B (SRB) sodium salt, RIPA (radio-Immunoprecipitation assay) buffer, and Tris-buffered saline with 0.1% Tween®-20 detergent were purchased from Sigma-Aldrich (Seoul, Korea). The LIVE/DEAD™ Viability/Cytotoxicity Kit, Alexa flour 488 goat antirabbit IgG, and SYTO 13 were purchased from Invitrogen (Carlsbad, CA, USA). The Bio-Rad Protein Assay was purchased from Bio-Rad Laboratories (Hercules, CA, USA). Advanced enhanced chemiluminescent (ECL) Western blotting detection reagents were purchased from Merck Millipore (Darmstadt, Germany).

### 2.2. Culture of Cells

Human squamous cell carcinoma lines from the tongue, SCC-25 cells, were cultivated in mixture of Dulbecco's modified Eagle's medium and F-12 (DMEM/F-12) with L-glutamine (4 mM), heat-inactivated 10% FBS, and penicillin/streptomycin (100 µg/mL). HaCaT cells (nonmalignant human keratinocytes) were cultured in DMEM. Both cells were cultured in a humidified incubator at 37 °C with a 5% $CO_2$ atmosphere.

### 2.3. Nonthermal Atmospheric Pressure Plasma Supplier

The NTP machine used in this study was developed by Feagle Company (Yangsan-si, Kyeongsangnam-do, Korea). The machine consisted of 1 dielectric and 2 electrodes. The inner electrode was composed of stainless steel with a 10-mm-wide alumina cylinder. The external electrode was wrapped with copper tape. Argon gas was passed between the internal electrode and the alumina cylinder at a flow rate of 2 standard L/min. The NTP was produced between the inner and outer electrodes, and the plasma flow temperature at the electrode was kept below 35 °C for 10 min. The ultraviolet rays emitted by the device were not detected by the ultraviolet (UV) sensor, but the plasma produced was mainly composed of OH radicals. Previous studies [23] have reported the comprehensive chemical characteristic of the NTP produced by this machine. To avoid indiscriminate cell death, the voltage was monitored and maintained at the corresponding value. Cells were cultured in Nunc™ Glass Bottom Dish (diameter, 40 mm) to treat SCC-25 or HaCaT cells ($0.5 \times 10^5$ cells/mL) with nonthermal plasma. The plate containing the cells was positioned under the end of the plasma jet. The distance between the cell and the machine was 10 mm. Plasma treatment was maintained for 5 min. A day prior to the plasma procedure, cells were cultivated in growth medium without or with PD-L1 antibody + GNP. Immediately prior to treatment, the dishes were rinsed with PBS to eliminate the unbound and nonselectively bound PD-L1 antibody + GNP (Figure 1) [26–28].

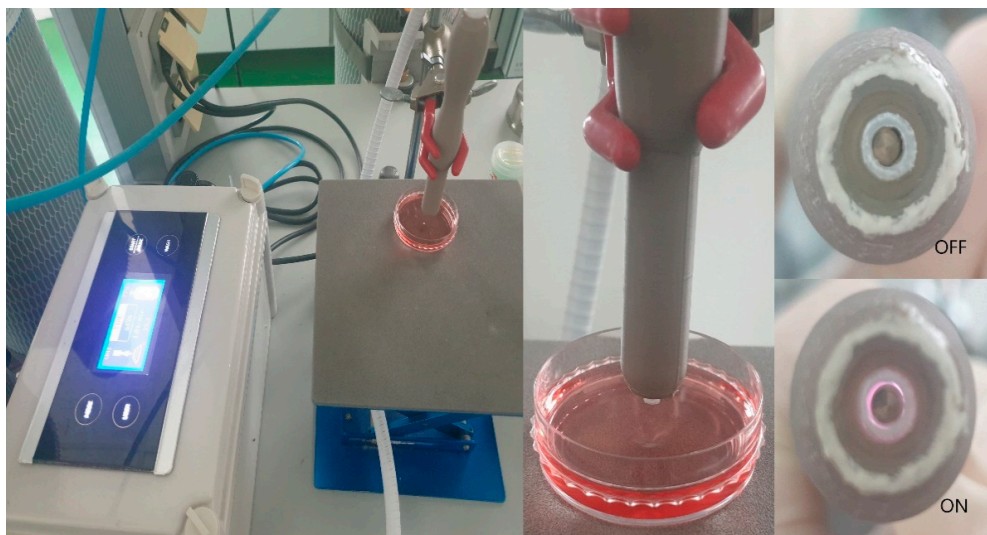

**Figure 1.** The figure of the NTP device used.

### 2.4. Preparation of PD-L1 Antibody-Conjugated Gold Nanoparticles

We mixed gold colloid suspension with 0.1-mg/mL 11-MUA solution and incubated it overnight to conjugate an antibody with GNP. After coating the surface of GNP with 11-MUA, it was reacted with 1-mM EDC as a crosslinker. Then, 1-mM NHS solution was added for 20 min at 4 °C to formulate an activated amine. Amine-activated GNPs were formed and coupled to the amine group on the PD-L1 antibody (1:10) and to the carboxyl groups on 11-MUA for 1 h at 4 °C with phosphate-buffered saline (PBS) (1 mM, pH 7.0) [23].

### 2.5. Cell Viability Assay

The SRB assay was used to evaluate the cell viability. Briefly, $0.5 \times 10^5$ cells/mL were seeded onto 24-well plates. After treatment with the GNP-PD-L1 antibody and a 2-h incubation period, SRB reagent (10 μL) was added to each well. After 2 h, the absorbance at 450 nm was measured using a microplate reader. This analysis was performed in triplicate.

### 2.6. LIVE/DEAD™ Viability/Cytotoxicity Assay

The LIVE/DEAD™ Viability/Cytotoxicity Kit was used to evaluate the cell membrane integrity and cytoplasmic function by assessment of the cytoplasmic esterase activity. This kit contains two fluorescent dyes. The first, calcein AM, passes through the cell membrane; after which, it is hydrolyzed by cytoplasmic esterase in living cells, appearing luminous at 515 nm (green color). The second, ethidium homodimer, fluoresces at an emission wavelength of 617 nm (red color) and binds to DNA in cells with disrupted cell membranes. This assay was performed using a Nunc™ Glass Bottom Dish. Briefly, cells $(0.5 \times 10^5$ cells/mL) were inoculated, conjugated for 2 h, and analyzed after NTP treatment and a 2-h incubation period. Cells were rinsed twice with PBS and examined using the LIVE/DEAD™ kit according to the product's protocols [29,30].

### 2.7. Immunocytochemistry

Cells were mixed with PD-L1 ab + GNP in a 1:1 ratio for 2 h. The treated cells were washed 3 times with PBS for 5 min and then fixed with 4% paraformaldehyde for 10 min. The cells were then blocked with 5% bovine serum albumin (BSA) in PBS for 1 h. After incubation for 1 h with Alexa Flour 488 goat antirabbit IgG at 37 °C, the cells were washed 3 times with PBS, followed by incubation with SYTO 13 for 10 min at 37 °C. After washing 3 times with PBS, the fluorescence staining was examined using a confocal microscope.

After the treatment of NTP or PD-L1 ab + GNP + NTP, the cells were fixed in 4% paraformaldehyde for 10 min. Fixed cells were incubated with AIF and cyt C antibody, respectively, for 1 h at 37 °C. After washing 3 times each with PBS, the cells were incubated

with Alexa Flour 488 and 594 antibodies at 37 °C for 1 h. Fluorescent images were observed and analyzed under a confocal microscope.

### 2.8. Western Blot Analysis

After treatment, the cells were cultivated for 4 h and then rinsed twice with PBS. Cells were harvested on ice after mixing with the RIPA buffer-containing phosphatase inhibitor. After collecting the lysates in a tube, the cells were centrifuged at 14,000 rpm for 20 min at 4 °C. The resulting supernatant was separated and used for the quantification of the lysates at 30 μg using the Bio-Rad Protein Assay. The solution was then boiled at 95 °C for 5 min. SDS-PAGE (sodium dodecyl sulfate polyacrylamide gel electrophoresis) (8–15% gel) was used to resolve the cell lysates (30 μg) and transfer them to the polyvinylidene difluoride (PVDF) membranes. After transference, the gel was blocked using 5% skim milk and Tris-buffered saline supplemented with 0.1% Tween®-20 detergent (20 mM-Tris, 150-mM NaCl, and 0.1% Tween-20) for 1 h. The membrane was incubated with antibodies against PD-L1 (1:500), PARP (1:1000), and cleaved caspase-3 (1:1000). After washing with PBS, the membrane was incubated with a labeled secondary antibody (1:500) for 2 h at room temperature. Advanced ECL Western blotting detection reagents were used to identify the bands. A β-actin antibody (1:1000) was used as the loading control.

### 2.9. Statistical Analysis

Each experiment was repeated in triplicate, with the standard deviations plotted as error bars. Experimental groups and a control group were compared using the paired *t*-test in Excel (Microsoft, Redmond, WA, USA). Statistical significance was determined as $p < 0.05$.

## 3. Results

### 3.1. Comparison of PD-L1 Expression between Oral Squamous Cell Carcinoma Cells and Nonmalignant Keratinocyte Cells

To confirm that PD-L1 is a selective marker for squamous cell carcinoma (SCC) cells, the expression of PD-L1 in SCC and HaCaT cells was confirmed by Western blotting. Higher levels of PD-L1 expression were observed in cancer cells compared to HaCaT cells (i.e., nonmalignant keratinocyte cells). Among the four oral SCC cell lines, SCC-25 cells expressed the highest levels of PD-L1 (Figure 2).

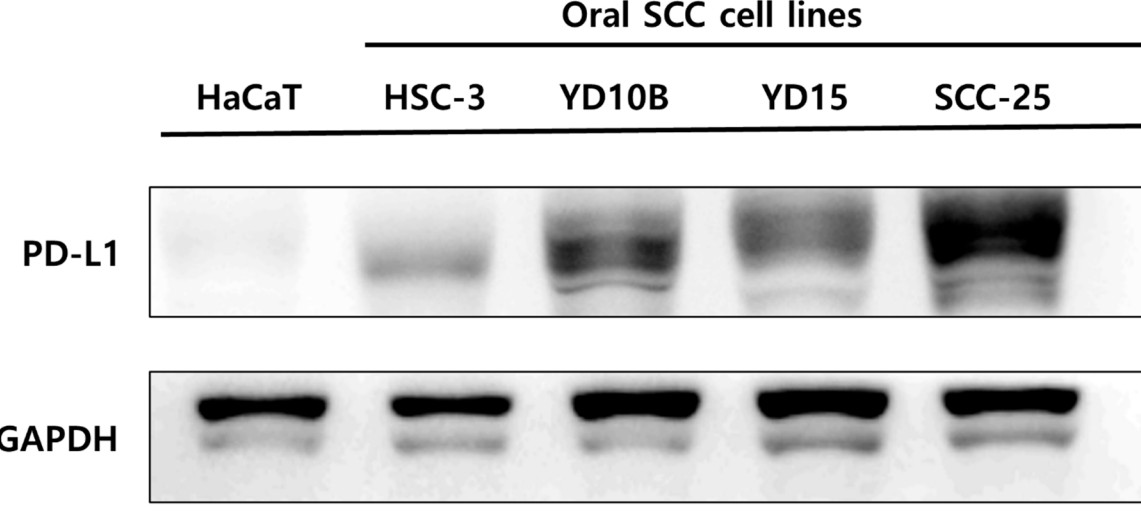

**Figure 2.** Western blot assay of the PD-L1 expression in oral squamous cell carcinoma cells. PD-L1 expression was the highest in SCC-25 cells.

### 3.2. The Binding of PD-L1 Antibody + GNP in SCC-25 Cells

An immunofluorescence analysis confirmed the binding of PD-L1 ab + GNP. As shown in Figure 3, PD-L1 ab + GNP stained red in SCC-25 cells after 2 h of treatment, primarily in the cytoplasmic area. In contrary, no localized and stained areas were observed in normal cells (HaCaT cells).

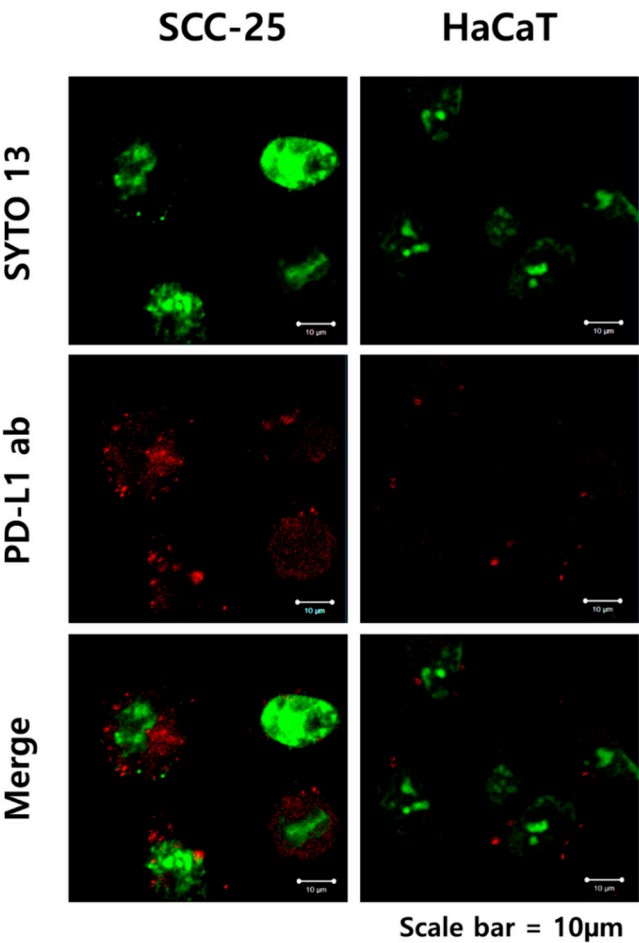

**Figure 3.** Cellular expression of PD-L1 and uptake of PD-L1 ab + GNP. Green spots represent a cell nucleus. Red spots represent PD-L1 ab + GNP bound on SCC-25 cells. Abbreviations: NT, nontreatment; ab, antibody; GNP, gold nanoparticle.

### 3.3. Selective Induction of Cancer Cell Death by PD-L1 ab + GNP and Nonthermal Plasma

The SRB analysis was performed to determine how PD-L1 ab + GNP and specific thermal plasma affect the survival of the SCC-25 and HaCaT cells. When the SCC-25 cells were processed with GNP, PD-L1 ab, or NTP, their viability changed to 92.59%, 110.05%, and 91.01%, respectively. However, the treatment had little effect on the survival rate of the HaCaT cells. When all three treatments were combined, the viability of the SCC-25 cells decreased by 59.96%, compared to that of HaCaT cells, which decreased to 88.14% (Figure 4).

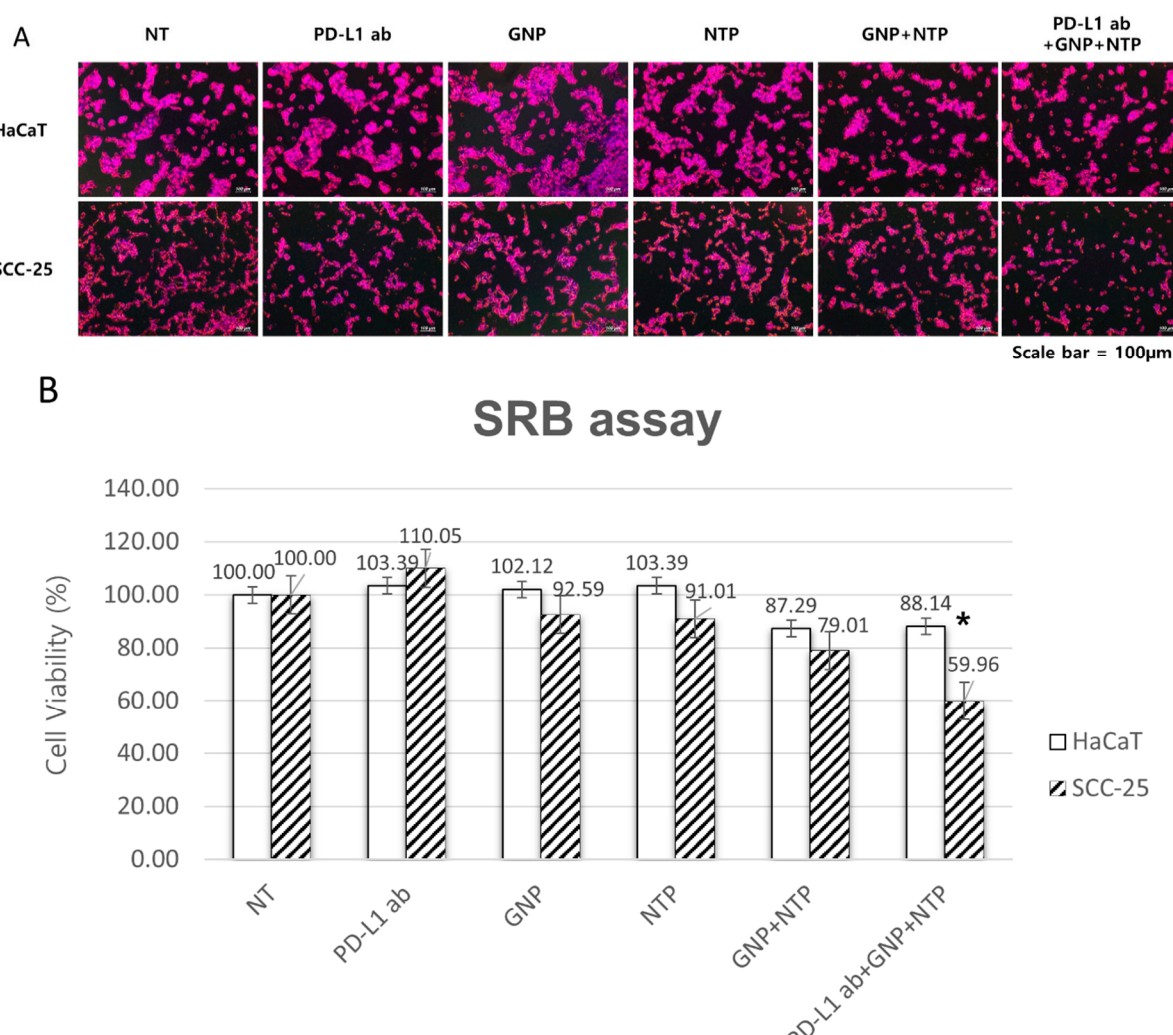

**Figure 4.** Effect of PD-L1 ab + GNP + NTP on the cell viability of SCC-25 cells and HaCaT cells. (**A**) According to the SRB assay, SCC-25 cells overall showed lower cell viability compared to HaCaT cells. Least number of cells were observed after PD-L1 ab + GNP + NTP treatment. (**B**) Data shown represent three independent experiments. * $p < 0.05$, statistical difference using a paired *t*-test. Abbreviations: NT, nontreatment; ab, antibody; GNP, gold nanoparticle; NTP, nonthermal plasma.

The LIVE/DEAD™ Viability/Cytotoxicity assay was used to evaluate the shape and viability of cells. Dead cells that emitted a red fluorescence once with ethidium bromide were bound to intracellular DNA. The treatment with PD-L1 ab + GNP + NTP significantly increased the number of dead cells in SCC-25 cells compared to the other treatments. In contrast, the survival rate of the HaCaT cells were barely affected by different combinations of treatments (Figure 5).

### 3.4. Induction of Apoptosis in SCC-25 Cells by the Treatment of PD-L1 ab + GNP + NTP

The Western blot analysis showed that the different treatment modalities displayed proteins that were typical apoptosis-inducing elements. Compared to the nontreatment group, the experimental groups showed higher levels of expression for apoptosis-related proteins, such as cleaved caspase-3 and cleaved PARP. Elevated expression levels of cleaved caspase-3 and cleaved PARP were observed in the PD-L1 ab + GNP + NTP group. β-actin served as the loading control. The experiments were repeated 3 times with similar results. The expression of cleaved caspase-3 and cleaved PARP was quantified by using Image J freeware from the National Institute of Health (Figure 6).

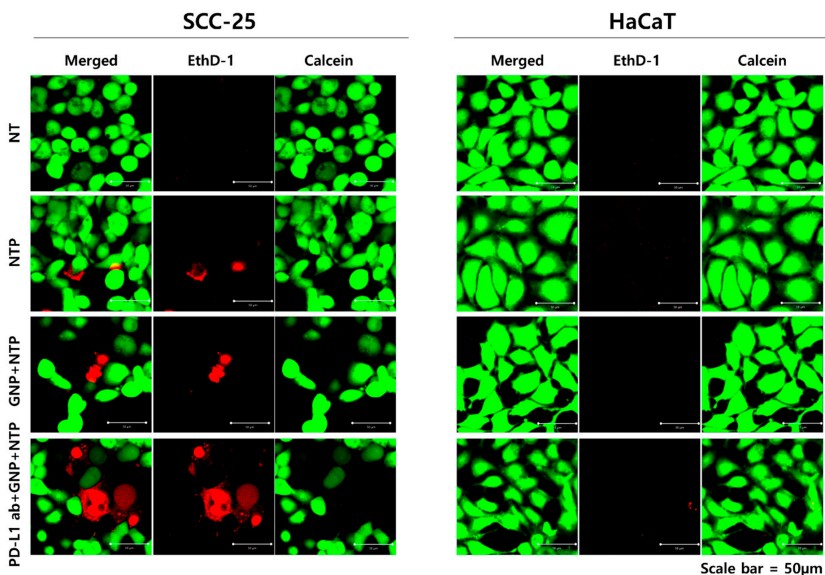

**Figure 5.** Comparison of the cell viability of SCC-25 and HaCaT cells following treatments using the LIVE/DEAD™ Viability/Cytotoxicity Kit. Green, live cells; red, dead cells. Compared to HaCaT cells, more dead cells were identified in SCC-25 cells among the experimental groups. Abbreviations: NT, nontreatment; ab, antibody; GNP, gold nanoparticle; NTP, nonthermal plasma.

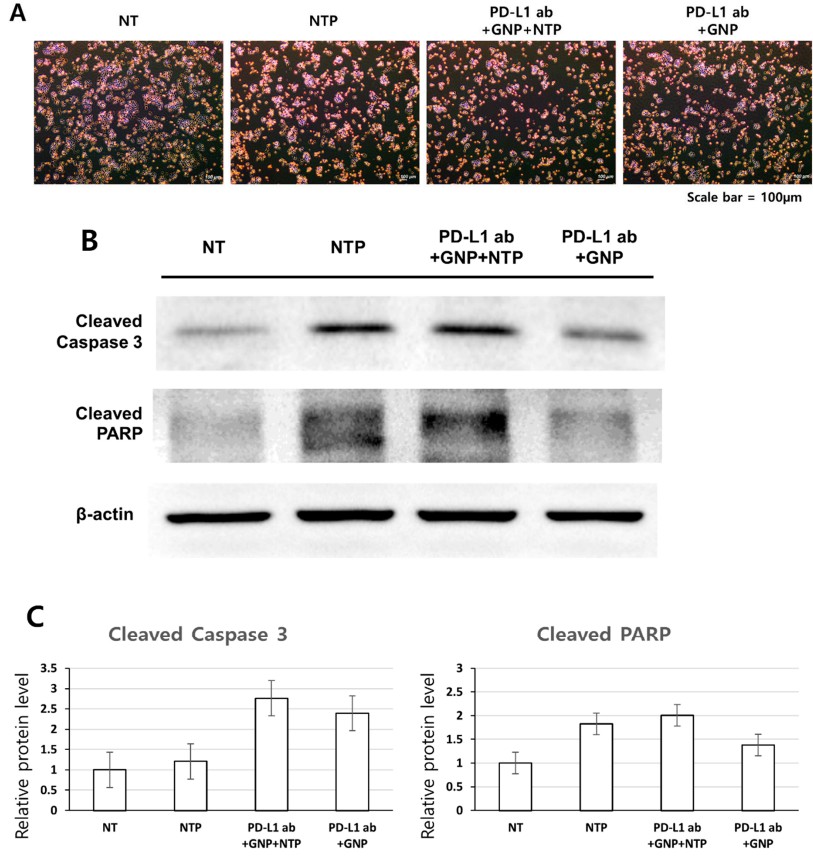

**Figure 6.** (**A**) Microscopic images. (**B**) Western blot analysis of cleaved caspase-3 and cleaved PARP. (**C**) Relative expression levels of cleaved caspase-3 and cleaved PARP protein in SCC-25 cells with NTP, PD-L1 ab + GNP + NTP and PD-L1 ab + GNP treatments. Abbreviations: NT, nontreatment; ab, antibody; GNP, gold nanoparticle; NTP, nonthermal plasma.

After the PD-L1 ab + GNP + NTP treatment, the mitochondrial apoptosis-related factors were assessed to verify the subcellular localization as evidence of apoptosis at the molecular level. The apoptosis-inducing factor (AIF) and cytochrome C (cyt C) in the control were clustered in punctuate distribution forms. On the contrary, in treated cells, those proteins were disseminated diffusely at immunofluorescence microscopy, demonstrating their release from the mitochondria. Especially, in the control cells, AIF was distributed mostly in the cytosol. On the other hand, AIF was detected in the nuclei of the treated cells (Figure 7).

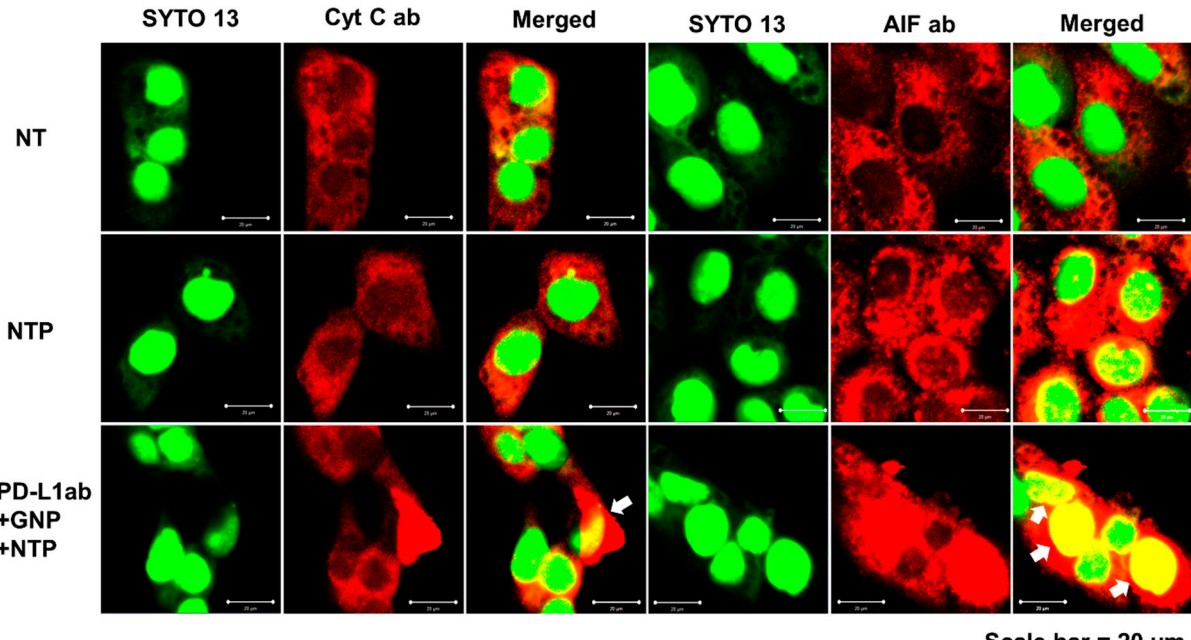

**Figure 7.** Induction of apoptosis in SCC-25 cells. Upon treatment with PD-L1 ab + GNP + NTP, after 4 h of incubation, immunocytochemistry visualized the redistribution of cyt C and AIF using an anti-cyt C antibody and an anti-AIF antibody, respectively. Translocation of cyt C and AIF are indicated with arrows. Abbreviations: NT, nontreatment; ab, antibody; GNP, gold nanoparticle; NTP, nonthermal plasma; cyt C, cytochrome C; AIF, apoptosis-inducing factor.

## 4. Discussion

In recent years, research on therapies using nanoparticles has gained popularity, with the aim of targeting cancer cells more selectively [28]. Strategies have included the combination of antibody and GNP conjugation increase delivery efficiency, diagnostic accuracy, and exert anticancer effects using lasers of specific wavelengths [27,29–32]. In the present study, NTP combined with Ab-conjugated GNP were used with an aim of increasing the targeting efficiency and enhancing cell death.

As a result, the combination of the PD-L1 antibody to GNP + NTP was found to exert a synergistic effect due to the antibody's targeting ability. According to the result of the SRB assay, this was the most effective when GNP, NTP, and PD-L1 ab were used in combination. Similar to the previous results [33], although the combination of GNP + NTP itself produced anticancer effects, the treatment of SCC-25 cells with PD-L1 ab + GNP + NTP was more effective. Thus, it was possible to implement a targeted treatment more effectively and selectively for cancer cells [23].

This study was designed based on the overexpression of PD-L1 in cancer cells. Although the PD-L1 antibody is widely used in immunotherapy [8], only a few studies deal with a combination of GNP and nonthermal plasma. Thus, this study aimed to establish a new combination approach for cancer treatment using antibodies against PD-L1. To this end, oral squamous cell carcinoma cells expressing high levels of PD-L1 were selected. Comparing the expression of PD-L1 between the SCC cells and HaCaT cells, a marked

increase in PD-L1 was observed in SCC cells, with the HaCaT cells weakly expressing PD-L1. According to our recent studies, the PD-L1 overexpression in oral cancer cells is associated with a poor prognosis. As such, SCC-25 cells, which showed high levels of PD-L1 expression, were selected as the primary cells for this experiment (Figure 2).

Immunofluorescence staining was performed to confirm the selective binding of the PD-L1 antibodies, which was only weakly observed in HaCaT cells. In contrast, SCC-25 cells treated with PD-L1 ab + GNP showed strong red staining in the cytoplasm around the nucleus (Figure 3). Similar to a previous study [34], higher levels of antibody-GNP uptake were observed in the cancer cells. These results confirmed the selective binding of SCC-25 cells by the PD-L1 antibody + GNP.

While the treatment with GNP and NTP had no effect on the HaCaT cells, a decrease in cell viability was observed in the SCC-25 cells. As demonstrated in previous studies [17,27], both GNP and NTP exert anticancer effects in cancer cells. When only PD-L1 ab was applied to SCC-25 cells, the cells proliferated to 110.1%, unlike normal cells. Further studies are required to determine why the cells proliferated after the application of the PD-L1 antibody alone. The survival of the SCC-25 cells treated with PD-L1 ab + GNP + NTP was markedly reduced compared to the HaCaT cells. This suggests that PD-L1 ab + GNP specifically attaches to the PD-L1 protein and synergizes with the nonthermal plasma (59.96% in SCC-25 cells), strongly inducing cell death. These results suggest that this combination method is effective for the selective treatment of cancer cells (Figure 4).

An analysis using a LIVE/DEAD™ Viability/Cytotoxicity Kit also showed similar results compared to the SRB assay. The former analysis enabled the differentiation between dead and living cells using different levels of fluorescent staining. Upon treatment with a combination of GNP + NTP and PD-L1 ab + GNP + NTP, higher levels of red cells (dead cells) were observed in SCC-25 cells compared to HaCaT cells (Figure 5).

The results of the Western blot analysis indicated that, compared to the nontreatment group, the treatment groups expressed higher levels of active apoptotic proteins—namely, cleaved caspase-3 and cleaved PARP (Figure 6), which indicates that the treatment with PD-L1 ab + GNP + NTP induced apoptosis in SCC-25 cells. In addition, cyt C and AIF are known as proteins that migrate from the mitochondria to the cytosol during apoptosis. Through immunofluorescence staining, cyt C and AIF migrated from the mitochondria to the cytoplasm and nucleus after the treatment of PD-L1 ab + GNP + NTP. Therefore, through these results, it can be assumed that the cell death is caused by apoptosis.

One of the limitations of the present study is the fact that PD-L1, which has an effect on immune cells, cannot be identified in relation to the immune system in the in vitro experiments. The PD-L1 protein was expressed on the surface of both cancer and immune cells. Additionally, NTP itself induced stimulation of the macrophage and immune responses [35,36]. Thus, upon the simultaneous application of the PD-L1 antibody, GNP, and NTP, in vivo experiments are needed to determine how each will affect the immune system of a living organism. Furthermore, although the PD-L1 antibody was used in immunotherapy, it has yet to be applied clinically in combination therapies with GNP and NTP. Thus, further research will be needed to verify the results presented in this study.

## 5. Conclusions

In this study, the therapeutic effects of a new combination of treatment approaches for oral squamous cell carcinoma were studied. Among the oral cancer cell lines, PD-L1 expression was highest in the SCC-25 cells. The selective binding of gold nanoparticles conjugated with PD-L1 antibody was greater in SCC-25 cells compared to HaCaT cells. A combination treatment using GNP conjugated with PD-L1 antibody and NTP markedly reduced the cell viability of SCC-25 cells and increased the expression of apoptosis-related proteins. In conclusion, the treatment with a combination of PD-L1 antibody, GNP, and NTP had a strong therapeutic effect on oral squamous cell carcinoma. Although further studies will be needed to confirm these therapeutic effects in vivo, this novel modality could lead the way for a new paradigm for the treatment of oral squamous cell carcinoma.

**Author Contributions:** Conceptualization, U.-K.K. and D.-S.H.; methodology, J.-H.C.; writing—original draft preparation, J.P.; writing—review and editing, U.-K.K., D.-S.H., G.-C.K., J.-H.B., and M.R.; visualization, Y.-S.J. and funding acquisition, U.-K.K. and D.-S.H. All authors have read and agreed to the published version of the manuscript.

**Funding:** This study was supported by National Research Foundation of Korea (NRF) grant funded by the Korea government (MIST) (NO.2019R1F1A1042528 and NO.2020R1A2C1100519).

**Institutional Review Board Statement:** Not applicable.

**Informed Consent Statement:** Not applicable.

**Data Availability Statement:** The data used to support the findings of this study are included within the article.

**Conflicts of Interest:** The authors declare that there are no conflicts of interest regarding the publication of this article.

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
