# Peer review of "Anticancer Efficacy of Nonthermal Plasma Therapy Combined with PD-L1 Antibody Conjugated Gold Nanoparticles on Oral Squamous Cell Carcinoma"

_applsci, doi:10.3390/app11104559_

Round 1

Reviewer 1 Report

Accept in present form

Author Response

Thank you for your acceptance.

Reviewer 2 Report

This paper by J. Park et al is interesting in that it studies the effect of the combination of PD-L1 antibody conjugated gold nanoparticles with non-thermal plasma at atmospheric pressure on oral squamous cancer cells, which, to my knowledge, has never been done before. It shows the beneficial effect of the combination of the treatments on the viability of cancer cells compared to separate treatments. The study is well conducted and the diagnostics are appropriate. English can be improved. Proofreading by an editor would be welcome. This article can be accepted after taking into account the following remarks:

- Abstract,

Lines 19 and 20: sentence too imprecise and general. We do not know what the authors are talking about. In addition, in current treatments there are already combination therapies, which leads to some confusion.

Lines 31 and 33: “among the different therapeutic approaches. "," Compared to other treatments. " must be specified. You might think that this is a comparison with other methods and not between the different treatments in the study themselves

- Introduction,

In general, beware of the use of the term "synergistic". Synergistic means that the result obtained with the combination of two actions is better than the sum of the expected results of the individual actions.

Line 41: "3600 people are diagnosed ..." it is necessary to specify in which country (Korea) it is, otherwise one can think that it is on a global scale.

Line 96: words are missing

Line 96-106 and references: The main point of the study is therapy combination with NTPs. This point is barely discussed and only introduced through a single sentence that specifies the study. Important references related to the use of plasmas for the treatment of cancer are missing. Considering only its topic, this paper should at least contain comments on, and refer to, the following papers:

H.R. Metelmann et al, Clinical experience with cold plasma in the treatment of locally advanced head and neck cancer, Clinical Plasma Medicine 2018, 9, 6-13 

R Guerrero-Preston et al, Cold atmospheric plasma treatment selectively targets head and neck squamous cell carcinoma cells,  International journal of molecular medicine, 2014, 34 (4), 941-946

MG Kong, M Keidar, and K Ostrikov, Plasmas meet nanoparticles—where synergies can advance the frontier of medicine, Journal of Physics D: Applied Physics, 2011, 44 (17), 17401

L. Brulle et al, Effects of a non thermal plasma treatment alone or in combination with gemcitabine in MIA PaCa2-luc orthotopic pancreatic carcinoma model, Plos one, 2012, 7 (12), e52653

In addition, reference 21, which can however be kept, is not cited in the right place. [21] relates to colorectal cancer cells but not to melanoma cells.

- Materials and Methods,

Paragraph 2.1.3: The plasma reactor is very poorly described. It is incomprehensible. It would be good to ask a plasma physicist to go over this paragraph.

There is an inconsistency between lines 140-141 and line 148. It is impossible to known if the authors are using or not a plasma jet. Lines 140-141, it is written: "did not expand like a plamsa jet" and line 148 "under the end of plasma jet" ...

- Results,

Line 242: The word "decreased" is not appropriate since for PD-L1 ab viability increases.

Lines 294-296: At the end of the caption of figure 6, there is a strange sentence that must date from a previous version of the paper.

-Discussion,

Lines 314-315: The sentence “Thus, this study… … against this protein.” is not correct and needs to be rephrased.

Lines 328-333: What is written in these few lines is not true. The effect of the combination of GNP and NTP is actually magnified in the case of HaCaT cells, but not in the case of SCC-25 cells. For these, the observed effect is simply additional. There is a synergistic effect only with the combined use of PD-L1 ab, GNP and NTP. This paragraph should therefore be reformulated.

Lines 346-353: Specifically for the results of this paragraph, but also for the whole study in general, it is a pity that the results of the association of PD-L1 ab and GNP only are not presented. This could have been interesting in particular in the detection of the levels of active apoptotic proteins and in the case of the viability of cancer cells. The authors should consider this and justify their approach.

Lines 354-361: The role of the immune system may be crucial in the use of combination therapy in vivo. The authors point this out, but do not refer to the work already done in the field with NTPs (as already mentioned above for lines 96-106). They must place this study in the general context of the data published in the literature, for example:

A Lin et al, Uniform nanosecond pulsed dielectric barrier discharge plasma enhances anti‐tumor effects by induction of immunogenic cell death in tumors and stimulation of macrophages, Plasma processes and polymers, 2015, 12 (12), 1392-1399

V Miller, A Lin, and A Fridman, Why target immune cells for plasma treatment of cancer, Plasma Chemistry and Plasma Processing, 2016, 36 (1), 259-268

Author Response

Response to Reviewer 2 Comments

For a better article publication, thank you for your sincere opinions. Thank you for your sincere comments for better article publishing. With your opinion, you have broadened my insights.

  1. - Abstract, Lines 19 and 20: sentence too imprecise and general. We do not know what the authors are talking about. In addition, in current treatments there are already combination therapies, which leads to some confusion.

Response 1 : I agree with you. As your comment, we revised the sentence.

“Combination therapies for the treatment of oral squamous cell carcinoma have been studied ex-tensively and represent a synergistic approach with better outcomes than monotherapy.”

  1. Lines 31 and 33: “among the different therapeutic approaches. "," Compared to other treatments. " must be specified. You might think that this is a comparison with other methods and not between the different treatments in the study themselves

Response 2 : I agree with you. As your comment, we revised the sentences.

“Among the experimental groups, the highest cell death was observed upon treatment with PD-L1 antibody + GNP + NTP. Following western blot analysis and immunofluorescence staining, the expression of apoptosis-related proteins was found to increase after treatment with PD-L1 antibody + GNP + NTP among the other experimental groups. In conclusion, the treatment of SCC-25 cells with PD-L1 antibody + GNP + NTP significantly increased the number of dead cells compared to other experimental groups.”

  1. Introduction, In general, beware of the use of the term "synergistic". Synergistic means that the result obtained with the combination of two actions is better than the sum of the expected results of the individual actions.

Response 3 : I think that the reduction of cancer cells and the increase of immune cells are synergistic effects. The anticancer effect will be significantly higher when both are exerted than each anticancer effect, and some papers (10. Li, XY et al) used the same expression. We kept synergistic for parts and changed some words.

“Previous studies have attempted to create a cooperative anti-cancer effect on melanoma cells by combining GNP with NTP. … These results indicate that NTP is an ideal auxiliary anti-cancer treatment method, whose combination with other treatments (chemotherapy and nanoparticle therapy) may regulate the micro-environment of cancer cells and inhibit cancer cell growth via cooperative action”

  1. Line 41: "3600 people are diagnosed ..." it is necessary to specify in which country (Korea) it is, otherwise one can think that it is on a global scale.

Response 4 : Thank you for your comment. I added “in south korea” in this sentence.

“In South Korea, every year, approximately 3,600 people are diagnosed with oral cancer, resulting in approximately 1,200 deaths. “

  1. Line 96: words are missing, Line 96-106 and references: The main point of the study is therapy combination with NTPs. This point is barely discussed and only introduced through a single sentence that specifies the study. Important references related to the use of plasmas for the treatment of cancer are missing. Considering only its topic, this paper should at least contain comments on, and refer to, the following papers:

Response 5 : Thank you for your recommendation of reference papers. I thoroughly read your recommended papers and revised this paragraph to contain these contents of the papers.

“ NTP itself works effectively on head and neck squamous cell carcinoma cells [20], and recently, it has been reported that it has a remission effect when applied clinically. [21]”

  1. In addition, reference 21, which can however be kept, is not cited in the right place. [21] relates to colorectal cancer cells but not to melanoma cells.

Response 6 : I revised this sentence.

“Also, previous studies have attempted to create a cooperative anti-cancer effect on cancer cells by combining GNP with NTP. [22-24]”

  1. Materials and Methods, Paragraph 2.1.3: The plasma reactor is very poorly described. It is incomprehensible. It would be good to ask a plasma physicist to go over this paragraph.

Response 7 : I added photo of NTP machine for easy understaning on manuscript as a figure 1.

  1. There is an inconsistency between lines 140-141 and line 148. It is impossible to known if the authors are using or not a plasma jet. Lines 140-141, it is written: "did not expand like a plamsa jet" and line 148 "under the end of plasma jet" ...

Response 8 : I revised this sentence. I removed "did not expand like a plamsa jet".

  1. Results, Line 242: The word "decreased" is not appropriate since for PD-L1 ab viability increases.

Response 9 : As your comment, we revised the sentences.

“When SCC-25 cells were processed with GNP, PD-L1 ab, or NTP, their viability was changed de-creased to 92.59%, 110.05%, and 91.01%, respectively.”

  1. Lines 294-296: At the end of the caption of figure 6, there is a strange sentence that must date from a previous version of the paper.

Response 10 : I apologize for this mistake. I removed it.

  1. Discussion, Lines 314-315: The sentence “Thus, this study… … against this protein.” is not correct and needs to be rephrased.

Response 11 : As your comment, we revised the sentences.

“Thus, this study aimed to establish a new combination treatment approach for cancer treatment using antibodies against this proteinPD-L1”

  1. Lines 328-333: What is written in these few lines is not true. The effect of the combination of GNP and NTP is actually magnified in the case of HaCaT cells, but not in the case of SCC-25 cells. For these, the observed effect is simply additional. There is a synergistic effect only with the combined use of PD-L1 ab, GNP and NTP. This paragraph should therefore be reformulated.

Response 12 : I agree with your comments. I deleted the misinterpreted part.

“While treatment with GNP and NTP had no effect on HaCaT cells, a decrease in cell viability was observed in SCC-25 cells. As demonstrated in previous studies [17,27], both GNP and NTP exert anti-cancer effects in cancer cells”

  1. Lines 346-353: Specifically for the results of this paragraph, but also for the whole study in general, it is a pity that the results of the association of PD-L1 ab and GNP only are not presented. This could have been interesting in particular in the detection of the levels of active apoptotic proteins and in the case of the viability of cancer cells. The authors should consider this and justify their approach.

Response 13: Thanks for your comment. Regarding this part, I am sorry too, and I wanted to proceed with the experiment. However, frankly, due to the lack of research funding, the use of PD-L1 antibody had to be minimized. Later, if additional experiments are possible, this is definitely something I want to do.

  1. Lines 354-361: The role of the immune system may be crucial in the use of combination therapy in vivo. The authors point this out, but do not refer to the work already done in the field with NTPs (as already mentioned above for lines 96-106). They must place this study in the general context of the data published in the literature, for example:

Response 14: Thank you for recommending a good reference for an area that I haven't considered. It seems that the reference was mainly focused on PD-L1 antibody and gold nanoparticles. Thank you for expanding my insights.

“Also, NTP itself induce stimulation of macrophage and immune responses. [35,36]”

Reviewer 3 Report

Park et al. address a very topical and clinically relevant issue with the combination of NTP and immunotherapeutic therapy. If NTP therapy alone shows very promising anticancer efficacy, it could be much more effective when combined with established conventional therapeutic approaches.

The combination of NTP with gold nanoparticles functionalized with PD-L1 antibodies demonstrated significant efficacy against an oral squamous cell carcinoma model. The findings are important and certainly lead to further combination therapies with NTP treatment, but some minor points need to be addressed in the manuscript.

The following are my comments:

Minor points

- Please correct the typos and misspellings (e.g. CO2).

- Some phrases in the text seem unwieldy; the script would certainly benefit from proofreading by a native speaker.

- The term 'normal keratinocytes' should be replaced by 'non-malignant keratinocytes' in my opinion.

- If the project is not already completed, it would be very good if at least the key experiment in Figure 3 would be repeated using another malignant cell line (e.g. YD15).

- Figure 3 shows very interesting data that I think could be discussed a bit more. Why does anti-PD-L1 alone not work, but gold nanoparticles do? What happens when the NTP treatment duration is increased, do you have any data or hypotheses on this? Is there any hypothesis on these synergistic effects you have shown?

Author Response

Thank you for your comments.

  1. Please correct the typos and misspellings (e.g. CO2). Some phrases in the text seem unwieldy; the script would certainly benefit from proofreading by a native speaker.

Response 1 : I corrected misspellings. Then, I requested English correction again to a native speaker.

  1. The term 'normal keratinocytes' should be replaced by 'non-malignant keratinocytes' in my opinion.

Response 2 : Thank you for your recommendation. I agree with you. I revised it as you commented.

  1. If the project is not already completed, it would be very good if at least the key experiment in Figure 3 would be repeated using another malignant cell line (e.g. YD15).

Response 3 : Thank you for suggesting. However, unfortunately, the project is already completed. It seems that further experiments will be difficult to proceed. I am sorry for that. In the future, when additional experiments are possible, I will try.

  1. Figure 3 shows very interesting data that I think could be discussed a bit more.
  • Why does anti-PD-L1 alone not work, but gold nanoparticles do?

Response 4-1) : According to Patil, M.P., Kim, GD. 2017, gold nanoparticles are a novel agent in cancer therapy and show aggregation (Cui et al. 2012) and size-dependent cytotoxic activity against different cancer cells (Pan et al. 2007; Cui et al. 2012) which also depends on the dose of nanoparticles (Raghunandan et al. 2011; Patil et al. 2016a). The AuNPs showed cytotoxic activity via ROS production (Parida et al. 2014), by DNA damage (Patil et al. 2016a; Jayaraj et al. 2014; Mishra et al. 2016), and activation caspase cascade of apoptosis and mitochondrial dysfunctioning (Tiloke et al. 2016; Jayaraj et al. 2014).

  • What happens when the NTP treatment duration is increased, do you have any data or hypotheses on this? Is there any hypothesis on these synergistic effects you have shown?

Response 4-2) : After 5 minutes, all cells were died even on non-malignant cells. So, if plasma is applied to treatment in the actual clinic, precise guidelines for intensity and duration are necessary.